# Conflict detection and resolution in macaque frontal eye fields

Tao Yao [1,2✉] & Wim Vanduffel [1,2,3,4✉]

Stimulus-induced conflicts in decision-making tasks produce both behavioral and neuronal congruency effects. However, how and when conflicts are detected and resolved at the neuronal level remains largely unclear. To address these issues, we recorded from single neurons in the frontal eye fields of two macaques performing a conflict task. Although the temporal dynamics of the neuronal congruency effects are independent of the specific task rules, they are substantially different in target- and distractor-encoding neurons. Conflicts were detected ~100 ms after the conflict-inducing cue (20–30 ms after the visual response), which is much faster than predicted based on human EEG results. This suggests that conflict detection relies on a fast mechanism in frontal eye fields. Resolving the conflict at the neuronal level, however, requires between <400 ms to ~1000 ms, and shows profound interindividual differences and depends on task rules, indicating that it is a more complex and top-down driven process. Our findings illuminate the neuronal mechanisms underlying decision-making when a conflict is present, a crucial cognitive process playing a role in basic survival and high-level cognitive functions.

[1] Department of Neurosciences, Laboratory of Neuro- and Psychophysiology, KU Leuven Medical School, Leuven 3000, Belgium. [2] Leuven Brain Institute, KU Leuven, Leuven 3000, Belgium. [3] Athinoula A. Martinos Center for Biomedical Imaging, Massachusetts General Hospital, Charlestown, MA 02129, USA. [4] Department of Radiology, Harvard Medical School, Boston, MA 02144, USA. ✉email: taoyao12@hotmail.com; WVanduffel@mgh.harvard.edu

In daily life, we continuously rely on cognitive control processes to optimally exploit our limited cognitive recourses. This includes the inhibition of automatic habitual responses, selective attention to task-relevant information and ignoring task-irrelevant items, and shifting between different tasks based on varying contexts and internal goals. Most often, we adapt our behavior by relying on task-relevant information[1,2]. Task-irrelevant information, however, may conflict with task-relevant information processing and impair task performance. In laboratory settings, "conflict tasks" are frequently used to investigate conflict processing. Well-known examples include the Stroop, Flanker, Simon, Wisconsin card sorting, pro/anti-saccade, and countermanding tasks[3–9]. Performing high conflict or incongruent trials results in worse performance and longer reaction times compared to low conflict or congruent ones - a behavioral phenomenon known as the *congruency effect*[10,11]. We recently reported a profound conflict signal in neurons of the frontal eye fields (FEF), which we referred to as the neuronal congruency effect (NCE)[12]. The NCE reflects the difference in neuronal response (at both single-cell and population levels) evoked by congruent and incongruent conditions. This signal provides a compelling neuronal mechanism for explaining the behavioral congruency effect.

Previous functional imaging and electrophysiological studies in humans and non-human primates suggest that fronto-parietal areas are involved in conflict processes[3,11,13–31]. Yet, the temporal dynamics of conflict processing in the brain, including conflict detection and resolution, is poorly understood. Currently, the "conflict monitoring" (CM) model is the most prevalent theory to explain conflict processing. This theory suggests that a conflict is detected in medial frontal cortex (i.e., anterior cingulate cortex, ACC) after which this information is transmitted to lateral prefrontal cortex (LPFC) to initiate a conflict resolution process[16,32–34]. This theory, however, has been challenged by other models suggesting that a conflict can be detected and resolved within LPFC without relying on a signal from ACC[35–39]. Yet, conclusive neuronal evidence supporting either of these theories is limited due to the low temporal and/or spatial resolution of imaging techniques and the inconsistent results from single-unit recordings in ACC[5,28,40–44]. Investigating the temporal dynamics of conflict processing and target selection signals will be important to understand conflict processing in the brain, but also validate different theoretical and computational models.

To this end, we analyzed in detail the time-courses of the NCE and target selection signals in the FEF to investigate how and when a conflict is detected and resolved in the prefrontal cortex. We find that the temporal dynamics of the NCEs are independent of specific task rules, however, they are substantially different in target- and distractor-encoding neurons. The time course of the NCE indicates that the conflict can be detected by FEF neurons ~100 ms after a conflict-inducing cue. This suggests that conflict detection relies on a fast mechanism in FEF. On the other hand, conflict resolution at the neuronal level requires much longer time and shows profound differences across individuals and task rules, suggesting that it is a more complex process. Our study will help bridge the gap between abundant human imaging results and computational models related to conflict processing with single-neuron data from non-human primates.

## Results

### The behavioral congruency effect and NCE.

Two subjects (*Macaca mulatta*, Monkey S and R) were trained to perform the rule-switching task (Fig. 1, methods). The subjects were required to covertly pay attention to a target and detect its dimming by pressing a button while ignoring dimming of potential distractors.

Target location was determined by a combination of a task rule (color or spatial rule) and a cue (pink or red) in each trial. A trial was considered congruent when the spatial location (spatial rule) and color (color rule) of the cue indicated the same target location (Fig. 1). In 29 recording sessions (12 in Monkey S, 17 in Monkey R), we found a significant behavioral congruency effect (Fig. 2a, b). The average performance was lower for incongruent than congruent conditions (Fig. 2a, monkey S: 80.6% vs. 93.8%, $t_{11} = 16.07$, $p = 5.5e-9$; monkey R: 81.6% vs. 95.9%, $t_{16} = 13.5$, $p = 3.7e-10$, two-tailed paired $t$-test). Also, the RTs for incongruent conditions were significantly longer than for congruent ones (Fig. 2b, monkey S: 353 ms vs. 349 ms; $t_{11} = 2.85$, $p = 0.016$; monkey R: 377 ms vs. 365 ms, $t_{16} = 7.1$, $p = 2.7e-6$, two-tailed paired $t$-test).

We recorded single unit activity in the FEF of two monkeys using 16-channel V-probes (Plexon). First, we plotted the average populational response of FEF visual neurons with contralateral RFs (Fig. 2c). In total, 248 post-hoc sorted single units (121 from monkey S, 127 from monkey R) were included in our analysis (Methods). All trials were categorized into four conditions based on conflict level (congruent: blue lines; incongruent: red lines) and target locations relative to the RFs. Please note that in the current study, the target- and distractor-decoding neurons are the same group of neurons in different experimental conditions (i.e., TarIn: target inside RF, "target-decoding" neurons, solid lines in Fig. 2c; DisIn: distractor inside RF, "distractor-encoding" neurons, dashed lines in Fig. 2c). As expected, these FEF neurons were activated by the appearance of the white peripheral stimuli inside their RF (indicated by the first vertical dashed line), and their responses remained the same for the four conditions until cue onset (the second vertical dashed line). Only after cue onset, the neuronal responses started to deviate according to the trial type.

Next, we calculated the NCE, i.e., the difference in response between congruent and incongruent trials, separately for target- and distractor -encoding neurons (Fig. 2d, gray and dark dashed lines, respectively). Profoundly different NCE dynamics occurred immediately after cue onset (Fig. 2d, after the second vertical dashed line). Specifically, distractor-encoding neurons started to respond less during congruent versus incongruent trials (i.e., they showed a negative NCE), which persisted afterward. On the other hand, target-encoding neurons responded higher during congruent compared to incongruent conditions (positive NCE) immediately after cue onset, yet the NCE turned negative approximately ~200 ms after cue onset, with a (partial) recovery after ~750 ms (Fig. 2d). Next, in order to investigate the temporal dynamics of NCE and conflict detection, we analyzed in detail the time course of the NCE for the two rules and for distractor- and target-encoding neurons, respectively.

### The NCE time-course of distractor-encoding neurons and conflict detection.

The average responses of distractor-encoding neurons for congruent spatial (solid lines) and color rule (dashed lines) trials are indistinguishable after cue offset (Fig. 3a). Distractor-encoding neurons show a similar persistent negative NCE for both rules (Fig. 3b). Note that only during incongruent color rule trials the cue was presented inside the neuron's RF (dashed orange line in Fig. 3a), while in the three other conditions, the cue was presented outside the RF. This explains the slightly faster response latency for incongruent color rule trials and a shorter NCE latency for color compared to spatial rule trials (Fig. 3c, d). Thus, the NCE latency for color rule trials reflect the visual response latency of a visual stimulus in the first place, while such stimulus-driven activity cannot explain the NCE latency for spatial rule trials. We quantified the NCE latencies and estimated

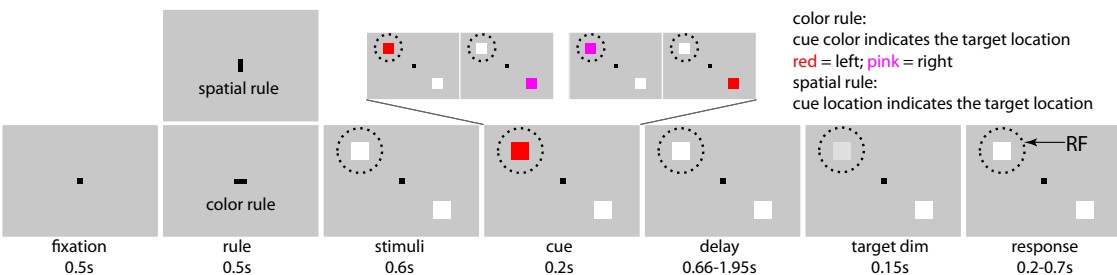

**Fig. 1 Task paradigm.** Subjects were asked to pay attention to a target stimulus and respond to its dimming. Trials were initiated after the subjects foveated a fixation point (FP) for 500 ms. The FP then turned to a horizontal or vertical bar (task rule: color or spatial rule). 500 ms later, the FP reappeared simultaneously with two peripheral stimuli (white squares). Next, one stimulus turned red or pink for 200 ms, and served as cue. In trials with horizontal bars (color rule), the color of the cue determined target location (red and pink indicating a target on the left and right, respectively), its spatial location being irrelevant. Conversely (vertical bar), the location of the cue indicated the target position, its color being irrelevant. Trials were subdivided into congruent (i.e., red presented on the left, and pink on right) and incongruent conditions (vice versa) based on the spatial location and color of the cue. Monkeys had to respond to target dimming by pressing a button with their left hand to receive a drop of fluid reward. To ensure that the monkeys covertly paid attention to the target, they had to foveate the FP during the entire trial while ignoring any distractor dimming, which occurred in 50% of the trials (before target dimming). All trial types were pseudo-randomly interleaved. The (virtual) dashed circle indicates the neuron's receptive field (RF).

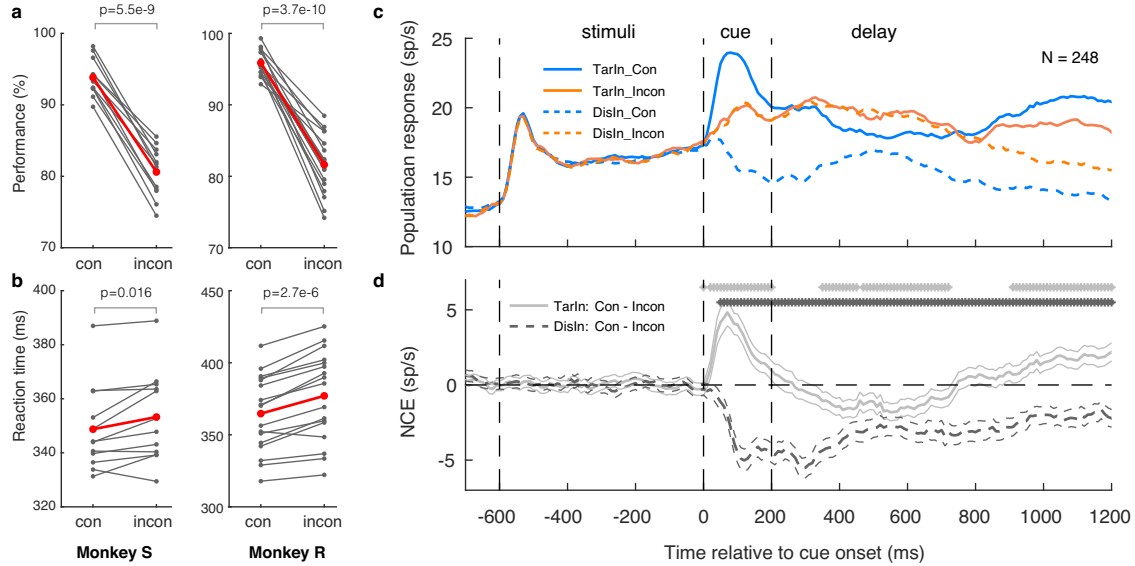

**Fig. 2 Behavioral and neuronal congruency effect.** Behavioral congruency effect. The performance accuracy was lower (**a**) and RTs were longer (**b**) for incongruent (incon) versus congruent (con) trials during the recording sessions (Monkey S/R: *n* = 12/17). Each pair of dots represented one recording session, the red pairs indicate the mean. **c** Population average peri-stimulus time histograms (PSTHs) from 248 neurons of 2 monkeys for the 4 conditions of the experiment, aligned to cue onset (0 ms). The neurons responded to the peripheral white stimuli (after the first dashed vertical line) and the responses were modulated by different cues (after the second dash vertical line). **d** The NCE time-course of target (gray solid lines) and distractor (black dashed lines) neurons. The thin lines indicate the s.e.m across neurons. The stars above the lines indicate the successive, moving 50 ms (stepped by 10 ms) time-bins in which the NCE differs significantly from zero (*p* < 0.05, Wilcoxon Signed Rank Test, WSRT): gray stars for target-encoding neurons and black stars for distractor-encoding neurons. The dashed vertical lines in **c** and **d** indicated the timing of (Left to right): white square stimuli onset, cue onset, and cue offset, respectively. TarIn/DisIn: conditions with target or distractor inside the RF, respectively.

their variability using a bootstrapping procedure separately for spatial and color rule trials and for both monkeys (see Methods, and Fig. 3d). The median/inter-quartile ranges (IQR) of the NCE latencies for spatial and color rule trials were 90 ms/20 ms and 70 ms/30 ms for monkey R, and 100 ms/20 ms and 70 ms/10 ms for monkey S. Thus, the median latencies were 20 ms (90–70 ms, Monkey R) to 30 ms (100–70 ms, Monkey S) faster for color compared to spatial rule trials, indicating that the NCE occurred 20–30 ms after the visual response in FEF. The relatively small IQRs indicate small variations of these latencies. The NCE's short latency shows that the distractor encoding-neurons can quickly distinguish congruent from incongruent cues. Thus, conflicts are detected quickly in the FEF, as suggested by the short and small variations of NCE latencies.

The PSTHs in Fig. 3a suggest that the responses of distractor-encoding neurons are suppressed in congruent conditions while increased in incongruent conditions after the cue compared to the response before the cue onset. To quantify this change, we calculated a modulation index (MI = $(R-R_b)/(R+R_b)$) comparing the response after the cue (100–300 ms after the cue, R) with a baseline response (200 ms preceding the cue, $R_b$) before the cue for each neuron (Fig. 3e–h). The results show that more distractor-encoding neurons have a negative MI, indicating suppressed responses after the cue, in congruent trials (Fig. 3e, g). More distractor-encoding neurons show a positive MI, hence increased responses after the cue, in incongruent trials (Fig. 3f, h). We found consistent results across monkeys (Supplementary Fig. 1). These results suggest that the conflict can be detected by

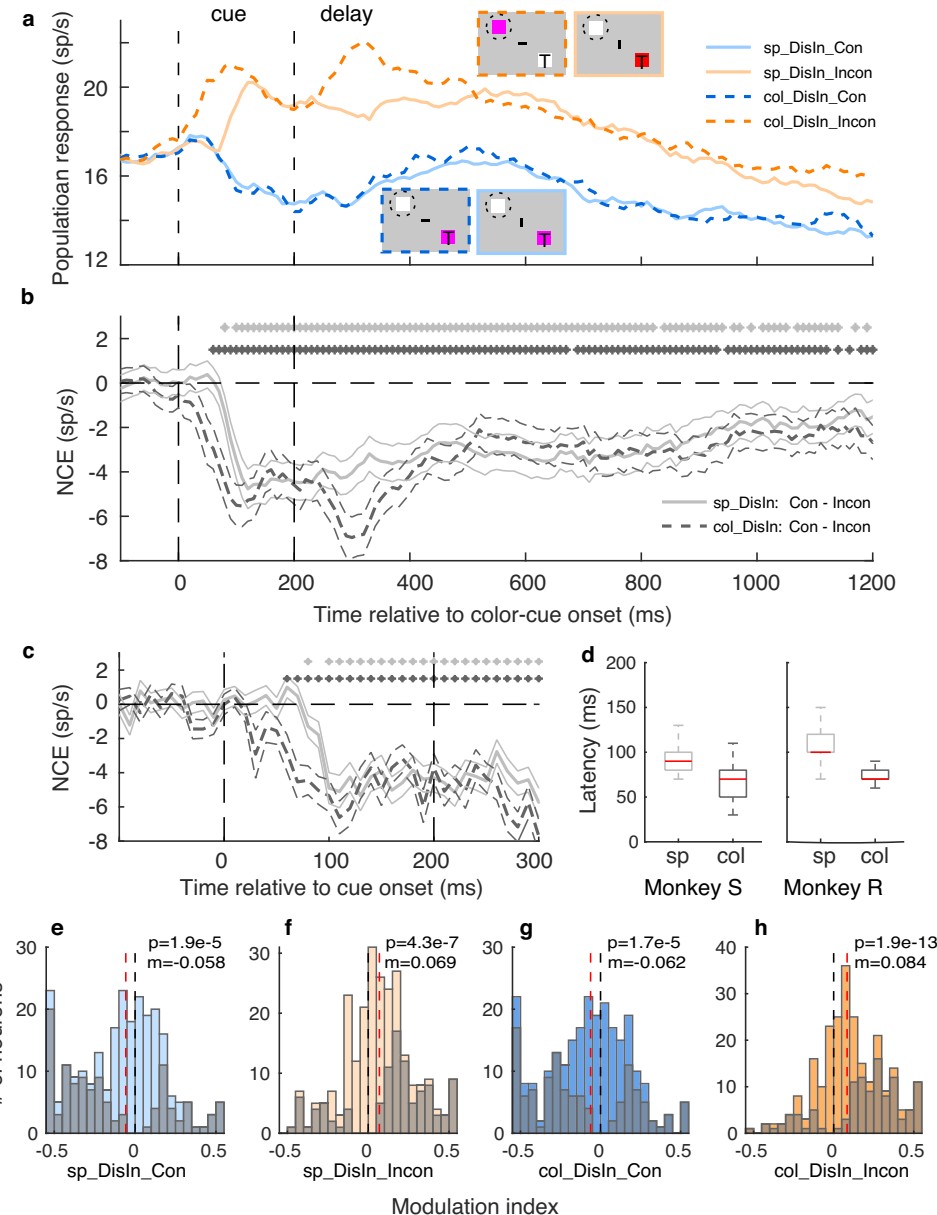

**Fig. 3 Time-course of the NCE for distractor-encoding neurons under two rules. a** Population average PSTHs for congruent (con, blue lines) and incongruent (incon, orange lines) trials either under spatial (sp, solid lines) or color (col, dashed lines) rules (see legends at top right, *n* = 248). The neuronal responses are higher in incongruent compared to congruent conditions for both rules. The "T", horizontal or vertical bar, and dashed circle in the insets indicate the target location, task rule, and RF in the corresponding condition, respectively. **b** The average response differences between congruent and incongruent conditions, i.e., the NCE. The thin lines indicate the s.e.m. across neurons. The stars indicate that the NCEs are significantly different from zero for a given moving 50 ms (stepped by 10 ms) time-bin (two-tailed WSRT, *p* < 0.05): gray and black stars for spatial and color rules, respectively. **c** Same as B, but NCEs for the color and spatial trails focus on a window between −100 and 300 ms relative to cue onset (=0 ms), and PSTHs are not smoothed. The two vertical dashed lines indicate cue onset and offset in **a–c**. **d** Boxplots of NCE latencies from two monkeys, with a bootstrapping method. The red lines indicate the median latencies, the bottom and top edges of the boxes indicate the 25th and 75th percentiles, respectively. The whiskers extend to the most extreme data points, which are not considered outliers. sp: spatial rule; col: color rule. **e–h** The response changes after the cue for distractor-encoding neurons for both the spatial and color rule. The modulation index (MI) was calculated as the difference between the average response (100–300 ms after the cue onset) and the baseline (200 ms preceding the cue) divided by their sum. The black vertical lines indicate zero change, the red vertical dashed lines indicate the medians for each condition. Negative MIs indicate suppressed responses after the cue compared to the baseline, while positive MIs indicate increased responses. The final bars on the two sides of the histogram sum all data values beyond −0.5 or 0.5. The *p* values (WSRT) indicate whether the medians of the MIs are significantly different from zero. The colors of the bars are matched to **a**. The gray bars indicate the number of neurons showing significant effects (*p* < 0.05, two-tailed WSRT).

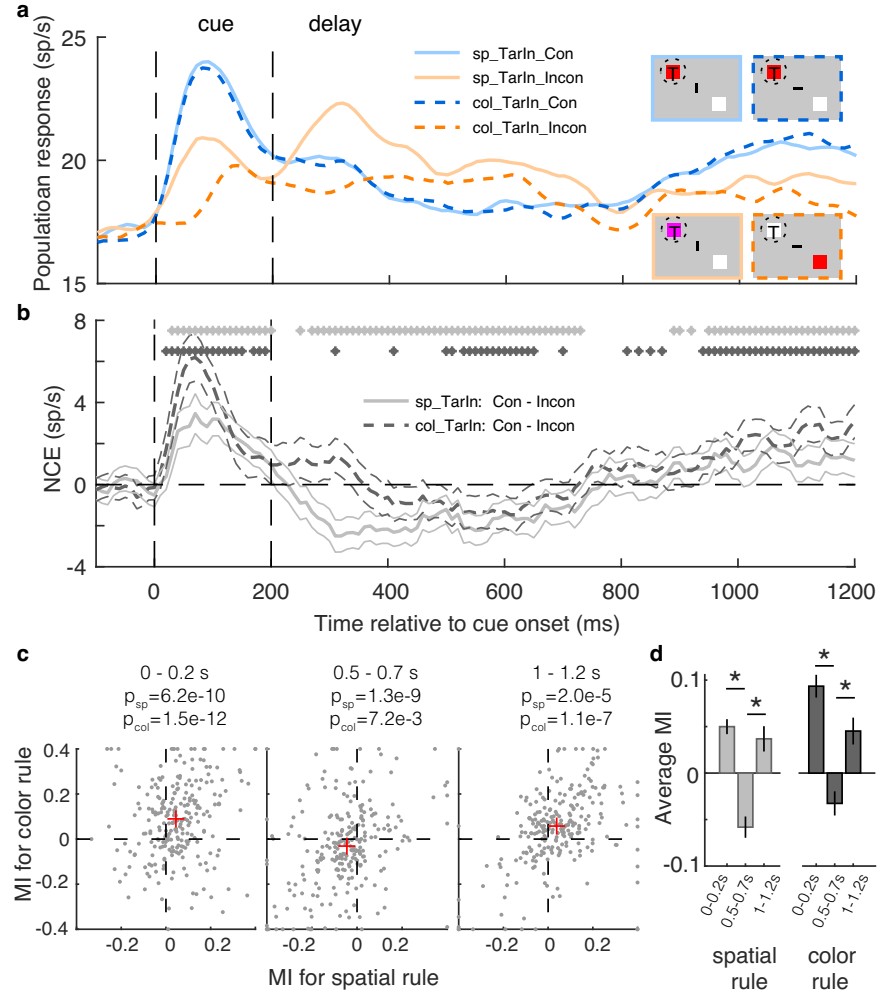

**Fig. 4 Time-course of NCE for target-encoding neurons under two rules. a, b** Same as Fig. 3a, b, but for target-encoding neurons (target inside the RF). **c** Scatter plots of MIs for color rule (y-axis) and spatial rule (x-axis) trials in time bins of 0–0.2 s (left), 0.5–0.7 s (middle), and 1–1.2 s (right) after cue onset. The red + indicates the median MIs. The p values (WSRT) show that the medians of the MIs are significantly different from zero for color ($p_{col}$) and spatial ($p_{sp}$) rule trials. MI = $(R_{con}-R_{incon})/(R_{con}+R_{incon})$. **d** The average MIs of the 248 neurons in 0–0.2 s and 1–1.2 s bins are significantly higher than the MI in the 0.5–0.7 s bin for both spatial (left) and color (right) trials. The error bars indicate the s.e.m. across neurons. *$p < 5e-5$ (2-tailed paired t-test).

distractor-encoding neurons by suppressing and increasing their response to the congruent and incongruent cues, respectively, even when the cues are not presented within their RFs (in trials with spatial rule).

**The NCE time-course of target-encoding neurons.** Consistent with the NCE of distractor-encoding neurons, spatial and color rule trials show similar NCE dynamics in target-encoding neurons (Fig. 4a, b). Except for the incongruent color rule trials (dashed orange line), the cues were always presented inside the RF, explaining the longer response latencies after cue onset in incongruent color rule trials (Fig. 4a). Since the latency of the NCE of target-encoding neurons might be contaminated by the visual difference between congruent and incongruent conditions, we refrained from performing a similar NCE latency analysis as for the distractor-encoding neurons. In line with Fig. 2d, we find a significant positive NCE peak immediately after cue onset, which turned negative after cue offset, to slowly recover after ~750 ms. To investigate the temporal dynamics of the NCE at single neuron level, we calculated a modulation index (MI) for each neuron in three bins after cue onset: 0–200 ms, 500–700 ms, and 1000–1200 ms. The MIs are calculated as the difference between the average response in the 200 ms bins for congruent and

incongruent trials divided by the sum of the two. Consistent with the NCE PSTHs, the NCE medians for the two task rules are significantly larger than zero in the 0–200 ms (median spatial/color MI: 0.04/0.09) and 1000–1200 ms bins (0.04/0.06) after cue onset, while negative in the 500–700 ms bin (−0.04/−0.03) (Fig. 4c). Further paired t-tests indicate that the mean MIs in the 0–200 ms and 1000–1200 ms bins are significantly higher than in the 500–700 ms bin for both spatial and color rule trials (Fig. 4d, all $p < 5e-5$), which is consistent with the PSTH (Fig. 4b). Consistent results are found across monkeys (Supplementary Figs. 2 and 3). These results suggest that the conflict could also be detected by target-encoding neurons by responding higher to congruent than incongruent cues immediately after cue onset. Combined with the data from Fig. 3, our results suggest that conflict detection in FEF might rely on attentional mechanisms (see discussion). Both target- and distractor-encoding neurons respond higher in incongruent than congruent trials shortly after cue offset (Fig. 4a, orange lines higher than blue lines), leading to a negative or non-significant (zero) NCE (Fig. 4b), which may be related to conflict resolution processes during this period in incongruent trials. Afterward, the NCE becomes positive again (~750–1000 ms after cue onset), contributing to the behavioral congruency effect.

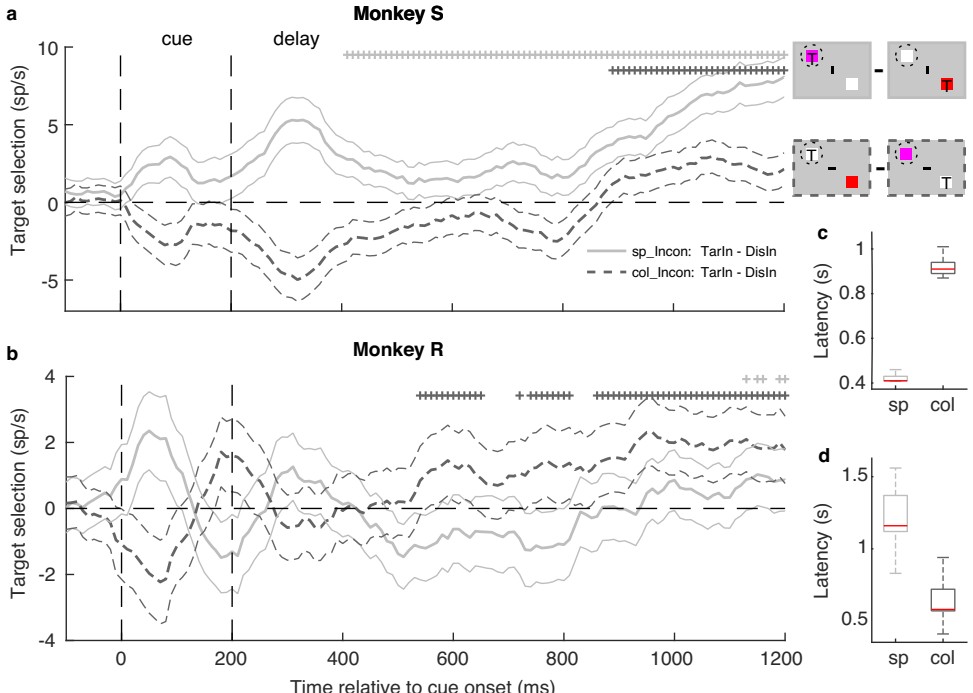

**Fig. 5 Time-course of conflict resolution under two rules. a**, **b** Population average of conflict resolution signals indexed by target selection (*y*-axis, response difference between target- and distractor-encoding neurons, or a selective spatial attention signal) for spatial (sp, light lines) and color rule (col, dark lines) trials for incongruent conditions in two monkeys. The thin lines indicate the s.e.m. across neurons. The stars indicate that the average response of target-encoding neurons is significantly higher than distractor-encoding neurons for a given moving 100 ms (stepped by 10 ms) time-bin ($p < 0.05$, WSRT): gray and black stars for spatial and color rules, respectively. **c**, **d** Boxplot of conflict resolution latencies for the two rules and two monkeys, using bootstrapping (Methods). The red lines indicate the median latencies, the bottom and top edges of the boxes indicate the 25th and 75th percentiles, respectively. The whiskers extend to the most extreme data points not considered outliers.

**The time-course of conflict resolution**. Since we only included hit trials in our analysis, it is reasonable to assume that the conflict was resolved at behavioral level in these trials. To investigate how the conflict was resolved at neuronal level, we analyzed the neuronal target selection signal in incongruent trials. If the neurons showed a significant positive target selection signal (response difference between the TarIn and DisIn trials, i.e., a spatial attention signal), we assume that the conflict was resolved at neuronal level in FEF. We calculated the latencies and their variations of target selection for different animals and rules by bootstrapping (Methods). Although we found highly consistent results for conflict detection across monkeys and rules, our data showed very different temporal dynamics of conflict resolution across individual animals and task rules (Fig. 5).

Specifically, in Monkey S, the latency of the conflict resolution for spatial rule trials is 410 ms (or earlier) with small variance (median/IQR: 410/20 ms). Note that we considered "resolution" latencies only starting 410 ms after the cue onset to avoid contamination by visual activity of the cue itself (see Methods). Hence, 410 ms was a very conservative estimation in our study. If we assume that the visual response to the cue offset terminates ~100 ms after cue offset, the latency of the conflict resolution for spatial rule trials would be ~300 ms since the target selection signal was already robust around this time (Fig. 5a, light gray lines). The conflict resolution latency in color rule trials was about 500 ms longer than for spatial rule trials (median/IQR: 910/50 ms, Fig. 5a, b). In general, the target selection signal was higher in spatial than color rule trials (Fig. 5a, gray line is above the black dashed line). Monkey R, on the other hand, showed conflict resolution latencies in spatial rule trials of ~1000 ms, with a relatively large variation (median/IQR: 1160/250 ms).

This latency was ~500 ms earlier for color rule trials (median/IQR: 580/150 ms, Fig. 5c, d). In general, target selection for this monkey was better in color than spatial rule trials (Fig. 5c). It is important to note that the monkeys were not required to resolve the conflict immediately after the cue, since there was a long delay between the cue onset and target dimming. In sum, we find that in some conditions a conflict can be solved in less than 400 ms after cue onset. However, in other instances, it requires more than 1 s. Hence, our results suggest that conflict resolution is a more complex process compared to conflict detection, and depends on individual differences in task-solving strategies, task rules, and possibly other cognitive processes.

**The time course of NCE and the overall target selection signal**. To investigate the relationship between the NCE (black and gray lines, for target- and distractor-encoding neurons, respectively, in Fig. 6a) and the overall target selection signal, we performed a linear correlation analysis, whereby the overall target selection signal was estimated as the response difference between all TarIn versus DisIn trials (regardless of congruency, dark solid lines, Fig. 6a). We found that the overall target selection signal is significantly correlated with the NCE of both target- (Fig. 6b) and distractor-encoding neurons (Fig. 6c) in the 0–1000 ms time window after cue offset. However, we found a significantly higher ($p < 0.001$) correlation between the target selection signal and the NCE for target ($r = 0.85$) than distractor-encoding neurons ($r = 0.42$). Thus, the NCE of target and distractor-encoding neurons explains 72% and 18% of the variance of the overall target selection signal, respectively. Similar results are found across monkeys (Supplementary Fig. 4).

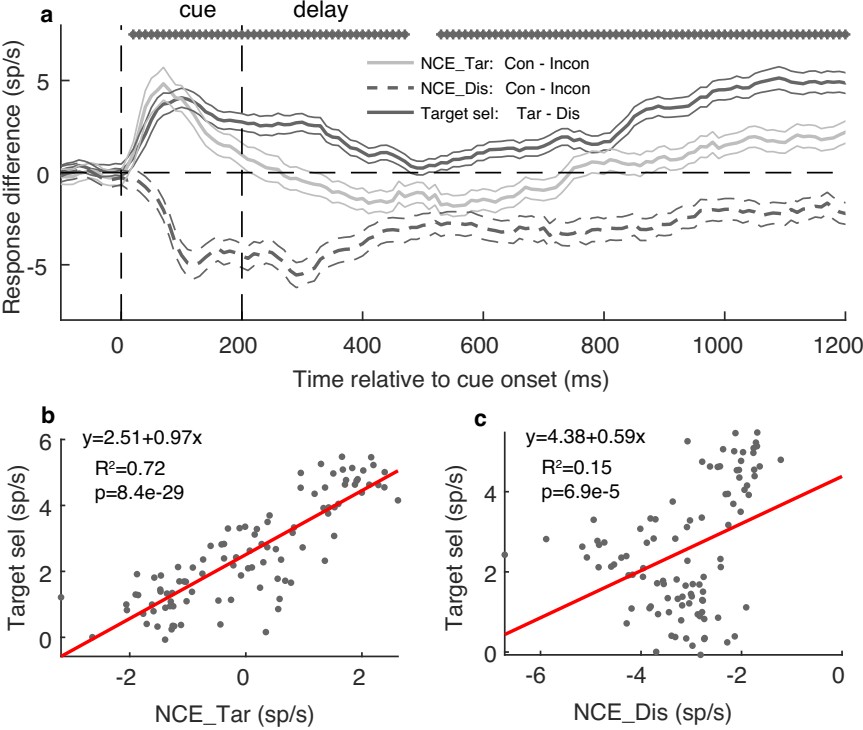

**Fig. 6 Time-course of the NCE and overall target selection signal. a** The time-course of the average attention effect (black line), and the NCE of target- (gray solid line) and distractor-encoding neurons (dashed lines). The thinner lines indicate the s.e.m across neurons. The stars indicate that the target selection signal is significantly different from zero for the given moving 50 ms (stepped by 10 ms) time bin. The linear correlation between the target selection signal (*y*-axis) and the NCE (*x*-axis) of target- (**b**) and distractor-encoding neurons (**c**). The dots represent the average NCE and target attention signal in a 10 ms non-smoothed time bin from 200 to 1200 ms after cue onset. The linear regression lines (red), $R^2$, and *p* values are shown. Note the higher $R^2$ and slope in **b** than **c**.

## Discussion

We report how neuronal congruency effects emerge and evolve in prefrontal cortex when monkeys perform a conflict task. The NCE of distractor-encoding neurons emerges surprisingly quickly (~100 ms) in the FEF, only about 20 to 30 ms after the first visual responses, and it persists until the end of the trial. The responses of distractor-encoding neurons are suppressed and enhanced during congruent and incongruent trials, respectively (Fig. 3). The NCE time-course of target-encoding neuron shows a U-shape after the conflict-inducing cue is presented (Fig. 4). Our data suggest that the conflict is detected fast and independent of individuals differences and task rules, while its resolution is more complex and may depend on interindividual task-solving strategies and other cognitive processes (Fig. 5).

We defined the NCE as the neuronal response differences between congruent and incongruent conditions[12], and the neuronal target selection signal as the response differences between target (TarIn) and distractor-encoding neurons (DisIn). By focusing on the time course of the NCE, we discovered that FEF neurons can distinguish congruent from incongruent cues 90–100 ms after the cue onset (on population level), which is only 20–30 ms after the visual response emerges in the FEF using our paradigm (Fig. 2). Thus, the FEF can detect a conflict within 100 ms. The conflict can be detected by both distractor- and target-encoding neurons. The former cells respond lower to congruent than incongruent cues, while the target-encoding neurons respond higher to congruent cues (Fig. 3). Such NCE closely resembles the neural response observed in the lateral intraparietal (LIP) area during anti/pro-saccades[45]. Specifically, LIP neurons exhibited a higher response in pro-saccade trials compared with anti-saccade trials when the saccade was directed to the neuron's responsive field. However, we cannot ascertain whether the activities in this study are identical to the NCE we reported in FEF due to differences in data analysis. In another study on anti/pro-saccades, similar results were reported by the authors. They identified a "paradoxical activity" in LIP visual neurons during anti-saccade trials when the saccade was directed into the responsive field, and the visual stimulus appeared at an opposing location[46]. The authors noted that the latency of this "paradoxical activity" was approximately 110 ms, closely aligning with the NCE latency or conflict detection time observed in our study. Consequently, there is a possibility that this "paradoxical activity" and the NCE might represent the same signal associated with the processing of conflict information. The timing of conflict detection is also consistent with previous results indicating that a target selection signal in FEF appears around 90 ms after stimulus onset in a search task[47], suggesting shared mechanisms of conflict detection and target selection, especially with proactive suppression of distractors[47–49]. In other words, the distractor is proactively suppressed before it impairs target selection at both neuronal and behavioral levels in congruent compared to incongruent trials. Moreover, this process may contribute to the detection of the conflict. However, based on the current results only, it is difficult to fully understand the exact relationship between conflict processing and target selection.

Previous EEG studies suggest that the conflict detection signal emerges at around 250–500 ms, depending on the conflict tasks[50–55]. Our results, however, reveal that the conflict can be detected much earlier in the brain. Possibly, EEG studies missed this early conflict detection signal because target- and distractor-encoding neurons respond in an opposite manner to congruent and incongruent stimuli -EEG cannot distinguish signals from

these two groups of neurons when they are not segregated. Hence, the early detection signals may be averaged out. On the other hand, both target- and distractor-encoding neurons responded higher in incongruent than congruent trials around 200–700 ms after cue onset in our results (Figs. 2–4), and such consistent responses can be detected by EEG. For example, the ERP component N450 from frontal regions (peaked at ~450–500 ms) differentiates congruent from incongruent conditions[50,56]. Since this N450 signal lags the (fast) conflict detection process in the present study, it may represent the initial phase of the conflict resolution process instead.

Our results are not consistent with the CM theory stating that conflict detection signals originate in ACC. Firstly, the conflict is already detected ~100 ms after the presentation of the conflict-inducing cue, and only 20–30 ms after the visual response in FEF. Yet, it has been shown that visual response latencies of ACC neurons are on average 36 ms slower than those of FEF neurons[57]. Although one could still argue that the fastest ACC neurons send conflict signals to the slowest FEF neurons, such a scenario is rather unlikely. Secondly, the conflict is detected by enhancing and suppressing target- and distractor-encoding neurons in FEF, respectively. Such neurons need to have a spatially well-defined receptive field, for which there is currently little evidence in the ACC. Instead, our results support input-driven attention theories for conflict detection[35–39]. Specifically, attention can be directed to the target location shortly after cue onset (within 100 ms) in congruent trials. In this case, responses of target-encoding neurons are rapidly enhanced while those of distractor-encoding neurons are suppressed by attentional mechanisms. However, in incongruent trials, the target is not immediately selected after cue onset. Hence, in this case, attentional deployment to target locations is delayed. This proposal is also supported by previous studies[58–61] which showed that target selection signals can affect neuronal activity within 100 ms. Moreover, target selection signals or attentional signals can rapidly transfer from one population of neurons to another one[62,63], which is required since these signals have to be transmitted from cue- to target-processing neurons, which are spatially segregated in some of the conditions in our task (e.g., across the left and right FEF).

We found highly similar temporal dynamics of conflict detection signals across individual monkeys, independent of the task rule. Surprisingly, however, FEF neurons show substantially different temporal dynamics of conflict resolution signals (target selection) across individuals and rules (Fig. 5). We noticed that in some conditions of our experiment, FEF can resolve a conflict in less than 400 ms, while in other conditions more than 1 s is required (Fig. 5). A possible explanation for such long conflict resolution latencies may be because we did not employ a reaction time task. Instead, there is a delay period of at least 860 ms between the onset of the conflict-inducing cue and go-cue (i.e., target dimming, Fig. 1). Note that this delay was required to investigate the NCE void of stimulus differences, which can be present during cue and dimming epochs of the trial. Therefore, the monkeys might trade accuracy for speed and resolve the conflict in some conditions. Similar to the conflict detection, the neuronal conflict resolution in incongruent trials can also be explained by attentional mechanisms as proposed by multiple conflict models[8,10,23,35,37–39,64–67]. At behavioral level, conflict resolution corresponds to the process whereby subjects correctly relocate attention to the target. At the neuronal level, conflict resolution can be implied by the response difference between the target and distractor. In other words, the conflict resolution indicates that attentional enhancement and suppression is correctly implemented to the target- and distractor-encoding neurons, respectively. Although both conflict detection and

resolution may be explained by attentional mechanisms, our recent paper suggested that at least the latest part of the NCE, cannot be *entirely* explained by the amplitude of the attentional modulation between congruent and incongruent trials. Indeed, the amplitude of the NCE just before target dimming does not correlate with reaction times -which are indicators of attentional intensity at behavioral level. Independent of the underlying mechanisms, our results reveal that conflict resolution is a more complex process than conflict detection, and is affected by the task rules, interindividual differences, and possibly other cognitive processes.

Visual, visuomotor, and motor neurons have been identified in the FEF[68–72]. In the current study, we analyzed neurons that exhibited visual responses to peripheral stimuli (methods), therefore, only the visual and visuomotor neurons were considered. Since FEF neurons are typically not modulated by the planning and execution of manual operant behaviors[73], and since our task did not involve saccades, we speculate that the motor components of motor and visuomotor neurons may not exhibit conflict-related responses in our experiment. However, if a task would require subjects to make saccades, we regard it plausible that motor and visuomotor neurons would also show conflict-related responses. In that case, we speculate that if a saccade would be directed towards the neuron's motor field, the neuron would be more active in congruent compared to incongruent trials. Conversely, if a saccade was directed away from the neurons' motor field, it would exhibit higher responses during incongruent compared to congruent trials. Further studies, however, are needed to investigate the involvement of FEF's motor neurons in conflict processing when other operant behaviors than hand responses are required.

In summary, our results help explain how conflicts are processed in FEF. *Step 1, conflict detection*: When a conflict-inducing cue is presented, the difference between congruent and incongruent input is detected within ~100 ms by evoking different responses in target- and distractor-encoding neurons within the FEF. *Step 2, conflict resolution and target selection*: If the input corresponds to an incongruent signal, the FEF is not able to immediately distinguish target from distractor (Fig. 2c, orange solid and dashed lines). However, the responses of both target and distractor-encoding neurons following an incongruent cue are higher than the responses of target-encoding neurons after a congruent cue (orange versus blue curves at ~250 ms after cue onset in Fig. 2c, see also Figs. 2d and 4). This may reflect a conflict resolution process in incongruent conditions. After the conflict is resolved in incongruent trials, a target selection signal starts to ramp until the target is selected (Fig. 4). *Step 3, target holding in short term memory*: Once the target has been selected, the FEF can retain this information until the end of the trial. Nevertheless, even after long delays, target selection in incongruent trials is inferior compared to congruent conditions as indicated by the NCE just prior target dimming, which results in a behavioral congruency effect[12].

## Methods
**Experimental model and subject details**. The subjects and basic methods of this study are the same our previous article, where the details can be found[12]. Below follows an extensive summary. Two adult male rhesus monkeys (*Macaca mulatta*, 6–8.5 kg, 8 and 10 years old during the period of recordings, respectively) participated in the current study. All experimental procedures and animal care were performed in accordance with the National Institute of Health's Guide for the care and use of laboratory animals, European legislation (Directive 2010/63/EU) and were approved by the Ethical Committee of KU Leuven. The animals

were socially group-housed in cages between 16 and 32 m3 equipped with enrichment devices (toys, woods, ropes, foraging devices etc.) at the primate facility of the KU Leuven Medical School. The animals were exposed to natural light and additional artificial light for 12 h every day. During the study, the animals had unrestricted access to food and daily access to restricted volumes of fruits and water. On training and experimental days, the animals were allowed unlimited access to fluid through their performance during the experiments. Using operant conditioning techniques with positive reinforcers, the animals received fluid rewards for every correctly performed trial. Throughout the study, the animals' psychological and veterinary welfare was monitored daily by the veterinarians, the animal facility staff and the lab's scientists, all specialized in working with non-human primates. The two animals were healthy at the conclusion of our study and were subsequently employed in other studies.

Each monkey was implanted with an MRI-compatible head holder to minimize head movements during the training and recording. One standard recording chamber was also implanted in each monkey above the right frontal cortex to allow access to FEF, with implantation locations chosen based on preceding MRI scans. The details of the implant surgery were previously described in ref. [74].

**Experimental design**. This study did not involve randomization or blinding. We did not estimate sample-size before carrying out the study. No data or subjects were excluded from the analysis.

**Setup**. The experiments were performed in a dimly lit room with the only source of light being the display monitor. A Dell 17 inches LCD monitor at a distance of 57 cm from the monkeys' eyes was used to display the visual stimulus at a refresh rate of 60 Hz and a spatial resolution of around 40 pixels per degree. The monkeys were seated in a sphinx position in a custom-made primate chair, typically used for fMRI experiments[72]. Stimulus presentation, reward delivery, electrophysiological and behavioral data collection was controlled by custom software controlled by custom-built hardware and Dell Windows computers. The exact timing of the stimulus onsets and offsets was monitored by a photocell attached to the bottom-right corner of the LCD monitor. Eye-positions were monitored by an Iscan (Iscan, MA, USA) Infrared corneal reflection system at 120 Hz.

Neuronal activity was recorded extracellularly with Plexon 16-chanel V-probes (Plexon Inc., TX, USA). The 16 recording sites were aligned in a row with 150 um inter-site spacing. The neuronal signal was filtered (300–1000 Hz), amplified, digitized, and stored with a TDT system (TDT Inc., TX, USA) with a 23 kHz sampling rate. All neuronal signals were recorded and stored for offline analyses. Offline spike sorting was performed with Plexon's Offline Sorter to isolate single and multi units. The FEF was identified by referencing the recordings to the structural MRI, in addition to the functional properties of the recorded neurons. Structurally, the recording sites, in the anterior bank of the arcuate sulcus, were localized with T1-weighted MRI imaging (TR = 2.5 s, TE = 4.35 ms, TI = 850 ms). Functionally, the saccade direction and spatial tuning of the neurons was visually inspected online. A site was considered to be within the FEF, only when the neurons from at least 1 channel showed clear direction tuning for saccades and spatial tuning for visual stimuli (in all our recording sessions included in current study, we actually observed that the neurons from multiple channels showed tuning to spatial location and saccade directions). By combining the structural and functional evidence, we are confident that the locations we recorded were in FEF.

**Behavioral tasks and stimuli**. Once the recording 16-channel-probe (Plexon V-probe) arrived at target depth and the neuronal signals were stabilized, the monkeys first performed a fixation task whereby they maintained fixation on a central black fixation point (0.2 by 0.2 degrees). We identified the RF of neurons recorded from several channels by briefly flashing (200 ms) a white square stimulus on the gray background at one of 25 locations (5 by 5 grid, covering 25 by 20 degrees of the visual field). The RF location was determined by online inspection and analysis of the neuronal responses to the flashed stimuli. We could not map the RFs for the neurons recorded at all channels, since we focused on just a few channels during the recording, and the amount of trials that the subjects could perform was limited per day (preventing us to carefully map all the RFs from all neurons on all channels). After mapping the RFs from several channels, we selected that location covering most of the RFs based on the mapping results, for placing the target and distractor during the main task. Next, we measured the neuron's saccade direction tuning by asking the monkeys to perform a visually guided saccade task from the center fixation point to a peripheral saccade target (7 visual degrees from the center fixation point). The saccade target was randomly picked from a set of 8 possible locations (evenly separated by 45° around a circle). We visually inspected the saccade direction tuning online. After identification of the location to position the stimuli, and when at least some of the neurons recorded on the 16 channels showed tuning for saccade directions, we switched to the main task and recorded neuronal activity without further moving the V-probe.

For the main experiment, the monkeys were trained to perform a task-switching paradigm (Fig. 1), where they were trained to pay covert attention to a target stimulus and respond to its dimming while ignoring a distractor stimulus. The monkeys initiated a trial by foveating a black fixation point (0.2 by 0.2 degrees) at the center of the screen. After 500 ms of fixation, the fixation point changed to a horizontal (0.4 by 0.2 degrees) or vertical (0.2 by 0.4 degrees) fixation bar which served as task rule-cue (color rule, or spatial rule, respectively) for the current trial. Accordingly, for a horizontal bar (color rule), the target stimulus was indicated by the color of the subsequently shown color-cue (which appeared 1100 ms after the task rule-cue, e.g., red or cyan for left, and pink or blue for right; please note that we used two pairs of color-cues (red-pink and cyan-blue) for monkey R (each recording session only used one pair), and only one pair (red-pink) was used for Monkey S). The spatial location of the color-cue was irrelevant in these trials. Alternatively, when a vertical bar appeared (spatial rule), the target stimulus was indicated by the spatial location of the subsequent color-cue, its color being irrelevant. The rule-cue was presented for 500 ms, then, the original squared fixation point returned together with a pair of white peripheral stimuli (1 by 1 degree). The two stimuli were positioned at equal eccentricities, one of them was presented at the location determined by the RF mapping task within most of the recorded neurons' RFs, the other one at 180 degrees from the former (thus the two stimuli were central symmetrical, if one stimulus was presented in the top left quadrant, the second appeared in the bottom right quadrant). This alignment would maximize the distance between the two stimuli and ensure that one of them was within the recorded neurons' RF, while the other one was out (in present study, the two stimuli were separated by at least 14 visual degrees). After a delay of 600 ms following stimuli onset, one of the two white squares turned into a color-cue (1 by 1 degree) for 200 ms. Combined with the rule-cue, the target location was indicated either by the color (color-rule: horizontal bar; red and pink indicated that the target would be located at the left and right, respectively), or the location of the color-cue independent of its color (spatial rule: vertical bar).

The monkeys had to respond to a brief (150 ms) dimming in luminance of the target by pressing a button with their left hand (within 200–700 ms after the dimming). Target dimming occurred between 660–1950 ms after color-cue offset in every trial. Moreover, to ensure that the subjects were attending to the target rather than responding to any dimming, the subjects had to ignore similar dimming of the distractor, which happened randomly in 50% of the trials, and never more than once in a trial. Distractor dimming occurred between 200–1500 ms after color-cue offset, with the additional requirement that it happened at least 300 ms before target dimming. This separation ensured that the monkeys' responses to the distractor dimming could be identified and distinguished from their responses to the target dimming. Trials terminated 700 ms after the target dimming, and the monkey received a drop of juice if the button had been correctly pressed during this period. Note that there was a target dimming in each trial, so the monkey was required to make an identical operant response in each trial in order to be rewarded, thus, we excluded a stimulus-response conflict. During the task, the background was always gray (RGB values: 70, 70, 70; 4 cd/m2); the fixation point and the task rule-cue (horizontal and vertical fixation point) were black (RGB: 0, 0, 0; 0.1 cd/m2); the squared stimuli were white (RGB: 255, 255, 255; 77 cd/m2); the dimming of the squared stimuli corresponded to a gray stimulus (RGB: 210, 210, 210; 51 cd/m2). Monkeys had to maintain fixation within a (virtual) squared window of 2.5–3 visual degrees centered around the fixation point until they received the reward. Please note that in the paradigm, between the congruent and incongruent conditions, we controlled (1) the visual input of the stimuli, by analyzing the neuronal response to exactly the same stimuli in the delay period; (2) the response of the subjects by asking the subjects performing the same response in all trials; (3) the allocation of spatial attention by requiring subjects to attend the target stimulus within or out of the neurons' RF.

**Statistics and reproducibility**. The data used in this article are partially overlapped with that in our previous article[12], we focused on a different analysis of the neuronal data with the goal to investigate: (1) the temporal dynamics of the NCE, and (2) how and when a stimulus-induced conflict is detected and resolved within the FEF. In the previous article, we focused on the mere existence of a NCE within the FEF (just prior the dimming of the target) and the relationship between the NCE and behavior. The data analysis was performed using MATLAB (MathWorks, MA, USA). We performed 12 recording sessions in Monkey S, and 17 recording sessions in Monkey R, and all the sessions included 32–40 correct trials for each condition. The correct trials (hits) corresponded to trials in which the button was pressed within 200–700 ms after target dimming. Incorrect trials included all false alarm trials (i.e., when the monkeys pressed the button at the wrong time) and missed trials (the monkeys did not press a button within the 200–700 ms response window after target dimming). All trials during which fixation was interrupted were excluded from the analysis. The performance of each session was defined as the number of correct trials divided by the sum of the number of correct and incorrect trials.

Neuronal activity was recorded from the FEF in the right hemisphere using Plexon's 16-channel V-probes. The spikes were offline sorted into single- and multi-units using Plexon's offline sorter. A total of 591 single neurons (267 from Monkey S, 324 from Monkey R) were isolated using offline sorting. Since our design required that one of the two stimuli should be presented in the neurons' RF, and not all of the neurons satisfied with this since: (1) some of the neurons were not visually driven (by these white squares) in FEF, (2) multiple neurons were recorded with

the probe at the same time, some of their RFs did not cover the target nor the distractor. Therefore, we first identified the visually-driven neurons that were activated by the white squared stimuli before the color cues. A neuron was qualified as visually-driven when it showed a significantly higher response in the 0–500 ms time window after the onset of the two stimuli onset compared to the response in 200–500 ms after onset of the fixation point (two tailed paired t-test, $p < 0.05$) across all correct trials. Next, for these visually-driven neurons, to determine their RF at the left or right visual hemifield, we analyzed the neuronal response induced by the target dimming. Since we recorded from the right FEF, we would expect most of the neurons' RFs were in the left visual hemifield[75]. We only included those neurons in our further analysis that showed a significantly higher response to the target dimming in the left compared to the right hemifield (50–200 ms after target dimming onset, two tailed paired t-test, $p < 0.05$). Therefore, the RFs of the selected neurons would cover targets presented in the left visual hemifield, while targets presented in the right visual field were outside the neuron's RFs, which was confirmed by the neuronal response in Fig. 4a, b in our previous paper[12]. We found a clear response to the target dimming when the target was supposed to be inside the RF (solid lines), while the response to the target dimming was not clear when the target was supposed to be out of the RF (dashed lines). Using the additional criteria, we were able to select 248 visual neurons with pronounced contralateral (left) RFs (121 from Monkey S, 127 from Monkey R) for further analysis.

Peri-stimulus time histograms (PSTHs) were calculated by smoothing the data with a moving 100 ms time bins (moving mean, step by 10 ms). The average activity of the congruent and incongruent conditions was first calculated across trials for each neuron. The average NCE was calculated as the average activity difference between the congruent and incongruent conditions for each neuron. Then, to display average activity (Figs. 2c, 3a, 4a) and NCE (Figs. 2d, 3b, 4b), PSTHs were obtained by averaging and smoothing (see above) the data across all neurons (except for the PSTH in Fig. 3c, which was not smoothed). The average target selection signal (Figs. 5, 6a) was calculated as the averaged activity difference between the TarIn and DisIn conditions for each neuron. To display the target selection signal, PSTHs were obtained by averaging and smoothing (see above) the data across all neurons. For the bin-by-bin statistical significance tests of the NCE and target selection signal, we performed a *Wilcoxon Signed Rank Test* across neurons for each given bin to test if the NCE or the target selection signals were significantly different from zero. To avoid that the transient response to the brief dimming of the distractor stimulus affected the results, we excluded the period of 0–200 ms following the distractor dimming onset from the PSTH and firing-rate calculations, for trials including a distractor dimming. The trials were terminated at target dimming onset to avoid the effect of target dimming on the PSTH.

To estimate the populational latency of the conflict detection and resolution (Figs. 3d and 5c, d), firstly, we calculated the NCE's time-course (non-smoothed) for each distractor neuron for both spatial and color rule trials. Then, we tested whether the population NCE was significantly different from zero for each non-smoothed 10 ms time-bin from −100 to 1600 ms relative to cue onset with a *Wilcoxon Signed Rank* across neurons. The NCE or the conflict detection latency was defined as the first of five continuous 10 ms time-bin showing a significant NCE. We calculated the populational latency for the two monkeys separately. To estimate of the variability of the latency, we used a bootstrap procedure where, for each of 10,000 bootstrap repetitions, we randomly sampled N (the number of neurons for each monkey, i.e., 121 for Monkey S, and 127 for R) neurons with replacement from the original dataset (which had N neurons).

We then calculated the latency for this simulated dataset. This process was repeated 10,000 times to create a bootstrap distribution of latencies, and the IQR was used to summarize the variability of the bootstrap distribution. The distributions are summarized in the boxplots in Fig. 3 and 5 excluding the outliers (the outliers: larger than q3 + 2.7 σ × (q3–q1) or less than q1–2.7 σ × (q3–q1)). We used a similar procedure to estimate the latency of the conflict resolution with two differences (Fig. 5c, d). Firstly, we calculated conflict resolution latency from 400–1600 ms relative to cue onset. The target selection signal in incongruent trials before 400 ms could be contaminated by the cue itself (i.e., a cue inside vs. outside the RFs). Since the cue was presented for 200 ms, and the visual latency of the FEF is ~60–100 ms, we discarded another 100 ms for the conflict resolution latencies, which allowed the neuronal activity to recover after cue offset. Therefore, the earliest conflict resolution latency we could estimate is 410 ms. Note that the actual conflict resolution latency may be earlier in some conditions. Yet, this is not relevant for the specific question we aimed to address: are there differences in conflict resolution latencies as a function of task rule and/or subject. Differences were detectable (see results) since they happen to appear after 410 ms. Secondly, since the conflict resolution was more variable than conflict detection, we smoothed the NCE time course with a moving mean method (smoothed in 100 ms and stepped by 10 ms).

**Reporting summary.** Further information on research design is available in the Nature Portfolio Reporting Summary linked to this article.

## Data availability

The behavioral and neuronal data that support the findings of this study are available on Zenodo with the identifier: https://doi.org/10.5281/zenodo.1032658[76]. The source data behind the graphs in the paper can be found in Supplementary Data 1.

## Code availability

The codes that produce the figures of this study are available on Zenodo with the identifier: https://doi.org/10.5281/zenodo.1032658[76].

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

## Acknowledgements
The authors thank C. Fransen, A. Coeman, P. Kayenbergh, I. Puttemans, C. Ulens, A. Hermans, G. Meulemans, M. Depaep, W. Depuydt, and S. Verstraeten for technical and administrative support. This work received funding from KU Leuven C14/21/111; Fonds Wetenschappelijk Onderzoek (FWO-Flanders) G0E0520N, G0C1920N; and the European Union's Horizon 2020 Framework Program for Research and Innovation under grant agreement no. 945539 (Human Brain Project SGA3) to W.V.; and Fonds Wetenschappelijk Onderzoek (FWO-Flanders) 1501320 N, 12W0919N to T.Y.; T.Y. is a postdoctoral fellow of the FWO- Flanders.

## Author contributions
Conceptualization, funding acquisition, writing: T.Y. and W.V.; Methodology, investigation, data analysis: T.Y.; Supervision: W.V.

## Competing interests
The authors declare no competing interests.
