## [Peer Review File · Communications Biology]

Reviewers' comments:

Reviewer #1 (Remarks to the Author):

In this paper entitled "Conflict detection and resolution in macaque frontal eye fields", Yao and Vanduffel describe the neuronal responses in the frontal eye field (FEF) to congruency effects. They show that conflict detection by FEF neurons is extremely fast (and faster than predicted on the basis of human EEG results), suggesting that conflict detection in the FEF is automatic and bottom-up driven. In contrast, they show that resolving the conflict at the neuronal level requires much more time, and is individual and task rule specific, suggesting a more complex top-down driven process. Overall, this is a straight forward study, relying on strong methodological and analysis grounds. I thus recommend it for publication in *Communications Biology*, once my comments below have been addressed. Please note that most of these comments do not question the presented analyses but mostly seek to enhance them and clarify some aspects of the data interpretation.

Major

1. The authors consider target-encoding and distractor encoding neuronal responses independently. How coordinated are the temporal dynamics of these neurons across sessions in the different epoch of interest highlighted by the authors in the target-encoding analysis? In other words, the authors overall suggest a sequential model between these two neuronal populations, but what about the possibility of a more complex coordination all throughout the trial?
2. Related to the previous question, the authors concentrate on correct trial analysis. It would be very informative to the reader to understand in appropriately selected miss trials what task encoding epoch mostly contributed to the failure: is it about failure in conflict resolution in target-encoding neurons or is it that conflict resolution fails due to cue encoding strength or to the encoding of conflict in the distractor-encoding neurons?
3. Conflict resolution timing is very different between task rules and also between animals. This is extremely interesting. Is there a correlation between RTs and single trial conflict resolution timing (or at least at the session level, to estimate this latter parameter over several neuronal time series). What if a time pressure were to be placed on response? The correlation between overall behavioral performance and neuronal conflict resolution for each rule 1/ speaks in favor of a causal relationship between the two, but also 2/ speaks in favor of the fact that each task rule is treated in both monkeys independently. However, how much of this is related to the learning history of the monkeys. How were they taught the task? Are we sure human would resolve such tasks in a similar manner -i.e. using the same neurocomputations?
4. The authors might consider discussing Di Bello et al., *Cerebral Cortex*, 2022. This study describes proactive and reactive distractor suppression mechanisms. Cosman et al. *Current Biol.* 2018 show that these suppressive signals arise in FEF before they arise in V1. These two studies are different from the present study in many ways, yet they also deal with the selection of relevant information and suppression of irrelevant information. Here, the authors show very fast conflict detection. Because conflict cannot arise reactively as it results from a learning process of cue significance, I would think that this encoding of conflict, although extremely fast, is of proactive nature, similar to the proactive suppression described by Di Bello et al. What are the thoughts of the authors on this? How much is overlearning contributing to the reported observations? Are the terms automatic and bottom-up driven most appropriate?
5. The authors perform a heavy selection on the neurons included in the present study (as described in detail in the material and methods section). While this is not a problem, it does raise in my mind the question of what the other cells are actually doing and how they are contributing -or not- to the computations described by the authors. I am not requesting additional analysis, but I think it would be useful to the readers to discuss this at some point.

Minor

1. Can you please indicate in figure 3efgh and corresponding S1 plots proportion of units in each bin

with statistically different response between baseline and post cue response in interval of interest? while at pop level, incongruency effects are clear, this is not the case at single cell level. Indeed although median of MI depart from 0, this is very subtle. How do you interpret this? How predictive is the modulation index of these distractor-neurons for spatial cue (cong or incongr) of the modulation index for color cue? In other words, is the distractor-population implementing spatial and color information along a mixed selectivity schema or along a schema of two independent populations. MI distributions speaks for former, which is what you would expect in the FEF.

2. Figure 4, please show histograms for individual MIs in supplementary figure in order to back up reported results on median MI distributions on page 11.

Reviewer #2 (Remarks to the Author):

Review – Conflict resolution in macaque frontal eye fields

Yao and Vanduffel

The manuscript by Yao and Vanduffel describes the results of a study in which the authors sought to investigate the twin processes of conflict detection and resolution in the macaque FEF.

Two rhesus monkeys were each trained to perform two tasks – one in which the identity of a colour cue instructed the spatial location of a forthcoming target dimming, to which the animals responded with a button press. In this task, the color cues were presented at spatial locations either congruent or incongruent with respect to the location indicated by the color. In a second task, the spatial location of the cue indicated the spatial location of the target dimming, and the colour of the cue was irrelevant. In this case, congruent and incongruent were defined based on the relationship between the cue location and the direction instructed on their identities in the colour task. The authors report the presence of behavioral and neuronal congruency effects -(NCE) – increases in RTs and decreases in performance in incongruent vs congruent trials, and a neuronal NCE – the difference in discharge rates between congruent and incongruent trials when either the target or distractor was present in the neurons' response field. They additionally report the timecourses of the discharge rates in all experimental conditions for two populations of FEF neurons – distractor encoding and target encoding. Generally speaking, activity for distractors was higher for incongruent than congruent trials, reflecting the presence of conflict. Conversely, for target-preferring neurons, activity was greater for targets during the cue period for congruent than incongruent trials, and this pattern subsequently reversed and then resolved during the delay period, which the authors ascribe to a cognitive process of conflict resolution. The authors thus argue that signals related both conflict detection and resolution are present in the FEF, and that their data are inconsistent with theories positing the ACC as a site of conflict detection, which is then signalled to prefrontal areas such as the FEF.

Overall, the task is well designed, and the animals behavior is well-characterized and consistent with a task engendering conflict. A sufficient number of trials and neuronal data were collected to answer the questions at hand and all analyses seem appropriate. I have some specific comments I would like address, listed below. I would be prepared to accept the manuscript if the following are addressed.

1. It seems as though the authors set up a bit of a straw man in the introduction with respect to the conflict monitoring (CM) hypothesis, which implicates the ACC in conflict detection and frontal areas in conflict resolution. In point of fact, while human fMRI studies have supported this hypothesis, there have not been any observations of conflict-related activity in ACC in studies recording single unit activity during tasks invoking conflict done more than 15 years ago (i.e. Nakamura, Ito). They do

acknowledge these other studies in the discussion section but the introduction should be more inclusive of the facts and relative novelty here. They additionally seem to focus exclusively on ruling out the ACC as a site of conflict detection and based on prior work as well as their latency analysis, this seems plausible. They then, at least to my reading, seem to make the claim that conflict in this case is detected in the ACC and not elsewhere. Although some conflicting evidence does exist (i.e. Nakamura), prior work has found evidence for conflict signals in the SEF (Stuphorn et al, 2000) which could be sent to the FEF with a latency consistent with their observations. Thus, conflict need not necessarily be detected specifically by the FEF but rather it could be receiving this signal from the SEF, after which the resolution process follows. This would also suggest that conflict signals in the FEF are not “bottom up” as the authors argue in the discussion section. This should be acknowledged.

2. I am somewhat confused by the separate characterization of “distractor preferring” and “target preferring” populations here and wonder how these were established. To my mind, the design here would be to investigate the congruent and incongruent conditions for both tasks, with either the target and distractor inside the rf (i.e. as done in search tasks, where all conditions are compared within the populations of neurons). Since FEF neurons typically show initially indiscriminate responses to identical visual stimuli, unless there is some bias induced by training (Bichot et al., 1996), I wonder how this is the case. Or perhaps this is an artifact of the fact that the authors could not optimize the response fields for all neurons recorded. Some comment to clarify this would be appreciated.

3. Have the authors investigated any sequential effects ? This might strengthen the findings of conflict signals and conflict resolution here if there are sufficient trials to do this. One might expect, for instance, that conflict would be highest for spatial trials immediately following a color trial, since they must now ignore color information and use the spatial location of the cues to facilitate detection of target dimming – especially in the incongruent. In this case, one might expect a higher NCE on these trials, relative to those on which the preceding trial was the spatial task. Just a suggestion but I would be interested to know if the authors have considered or tried this.

4. The authors show evidence for behavioral and neuronal congruency effects (NCE). The implicit assumption here is that the NCE underlies the behavioral effect, but no relationship between the NCE and performance is described. Presumably the magnitude of the NCE in the last few hundred ms of the delay period reflects the extent to which the earlier conflict in the task has been resolved, and spatial attention deployed to the target and away from the distractor location. What if the authors were to perform a median split of the data based on the distributions of the NCEs for all of the target and distractor neurons in an epoch consisting of the last few hundred ms of the delay period, and then compare the performance for the high NCE vs low NCE trials ? Such an analysis could be most illuminating with respect to the relationship between the extent of conflict resolution and performance and provide an explicit link between the neural and behavioral data.

Minor points:

Line 93, performance reported for Monkey S does not match figure 2A.

Reviewers' comments:

Reviewer #1 (Remarks to the Author):

In this paper entitled “Conflict detection and resolution in macaque frontal eye fields”, Yao and Vanduffel describe the neuronal responses in the frontal eye field (FEF) to congruency effects. They show that conflict detection by FEF neurons is extremely fast (and faster than predicted on the basis of human EEG results), suggesting that conflict detection in the FEF is automatic and bottom-up driven. In contrast, they show that resolving the conflict at the neuronal level requires much more time, and is individual and task rule specific, suggesting a more complex top-down driven process. Overall, this is a straight forward study, relying on strong methodological and analysis grounds. I thus recommend it for publication in Communications Biology, once my comments below have been addressed. Please note that most of these comments do not question the presented analyses but mostly seek to enhance them and clarify some aspects of the data interpretation.

We thank the reviewer for the positive comments.

Major

1. The authors consider target-encoding and distractor encoding neuronal responses independently. How coordinated are the temporal dynamics of these neurons across sessions in the different epoch of interest highlighted by the authors in the target-encoding analysis? In other words, the authors overall suggest a sequential model between these two neuronal populations, but what about the possibility of a more complex coordination all throughout the trial?

We thank the reviewer for the suggestions. We agree that our proposed model pertains to conflict processing in the context of the conflict monitoring theory. The overall neural temporal dynamics can be described as follows: conflict detection

precedes conflict resolution, and after successful conflict resolution, the target can be accurately selected. In our study, conflict detection is determined by the appearance of an NCE (i.e., the response difference between congruent and incongruent trials). Conflict resolution is determined by the target selection signal (the response difference between target- and distractor-encoding neurons). The failure of conflict resolution may result in unsuccessful target selection. Therefore, in our study, we do not expect a conflict-related neuronal response (or an NCE) before the color cue in target- and distractor-encoding neurons. This is supported by Figure RR1: the neurons responded exactly the same before the color cue in all four conditions before the cue (Figure RR1A), suggesting that there is no target selection and conflict-related (Figure RR1B) signal before the color cue.

Figure RR1. Neuronal population response before the cue onset. The legend is the same as Figure 2CD, except for the alignment relative to rule onset. Please note that the neuronal responses are indistinguishable across conditions (A), therefore, the NCE is close to zero (B) before cue onset.

2. Related to the previous question, the authors concentrate on correct trial analysis. It would be very informative to the reader to understand in appropriately selected miss

trials what task encoding epoch mostly contributed to the failure: is it about failure in conflict resolution in target-encoding neurons or is it that conflict resolution fails due to cue encoding strength or to the encoding of conflict in the distractor-encoding neurons?

We agree with the reviewer that we mainly focused on the correct trials since we suppose that conflict is resolved only in the correct trials, and we assume that conflict resolution fails in error trials, resulting in failed target selection. To confirm this argument, we analyzed the temporal dynamics of the neuronal response in miss and false alarm incongruent trials (high conflict) in both target- and distractor-encoding neurons. Note that there were much fewer misses and false alarms than correct target selections (Figure 2A). As indicated in Figure-RR2, we found that the response to the target and distractor was undistinguishable after the cue (Figure-RR2A, blue vs. red lines), and the target selection signal is close to zero or even below zero (Figure-RR2B) indicating that the target was not correctly selected in FEF, which suggests that the conflict is not resolved in miss and false alarm trials. On the other hand, in the incongruent hit trials, we found the response of the target-encoding neurons responded significantly higher than the distractor-encoding neurons around 600ms after the cue (Figure-RR3A), resulting in a significant target selection signal (Figure-RR3B and Figure 5). In the current study, we cannot explain why conflict resolution fails in some trials. We only suggest that the lack of target selection signals in FEF (during miss trials) is a result of failed conflict resolution. To resolve this issue, further investigations are required.

Figure RR2. Neuronal response and target selection in incongruent error trials.

(A) The neuronal population responses in the incongruent trials under both rules. (B) The target selection signal in the incongruent error trials. Please note the small but significant signals around 100ms and 300ms after the cue mainly reflect the cue onset and offset respectively, and the target selection signal is close to zero after the cue offset response.

Figure RR3. Neuronal response and target selection in incongruent hit trials. (A) The neuronal population responses in the incongruent trials under both rules. **(B)** The target selection signal in the incongruent hit trials. As in Figure RR2 and Figure 5, we found the response related to cue onset and offset in hit trials as well. However, the target selection signal appears clearly only 800ms after the cue onset. Please note that Figures RR2 and RR3 combine the data from two monkeys. Please refer to Figure 5 for the individual results.

3. Conflict resolution timing is very different between task rules and also between animals. This is extremely interesting. Is there a correlation between RTs and single trial conflict resolution timing (or at least at the session level, to estimate this latter parameter over several neuronal time series)? What if a time pressure were to be placed on response? The correlation between overall behavioral performance and neuronal conflict resolution for each rule 1/ speaks in favor of a causal relationship between the two, but also 2/ speaks in favor of the fact that each task rule is treated in both monkeys independently. However, how much of this is related to the learning

history of the monkeys. How were they taught the task? Are we sure human would resolve such tasks in a similar manner -i.e. using the same neurocomputations?

It is a very good suggestion to investigate the correlation between the RTs and conflict resolution timing at the single neuron level. Unfortunately, this analysis is impossible in the current study, since, firstly, calculating the conflict resolution or target selection timing on single trial requires the recording of target- and distractor-encoding neurons simultaneously (i.e., in the left *and* right FEF), which was not done in the current study. Secondly, as indicated by the reviewer and discussed in our manuscript, we did not put time pressure on the animal to find the target after the cue (i.e. this was not a reaction time task to investigate the conflict resolution). Instead, there is a long delay between cue and response, which could be the reason that the conflict resolution or target selection processes took so long in some conditions in the current study. Therefore, the reaction time in our study does not fully reflect the duration of the conflict resolution, it only indicates the consequence of the conflict resolution: if the animals responded to the target dimming, it is reasonable to assume that the conflict was resolved successfully. Therefore, the RT and neuronal responses do not necessarily correspond to conflict resolution on behavioral and neuronal levels in our study, i.e., the correlation between them is not necessarily related to conflict processing. In our recently published study, we specifically investigated this correlation in detail (Yao and Vanduffel, 2023, *Cell Reports*).

A new reaction time experiment (e.g., asking the animals to make a response as soon as possible after the cue) would be an excellent follow-up study, however, beyond the reach of the present manuscript. In this hypothetical new experiment, RTs would not only reflect the consequence of the conflict resolution but also when it is implemented. Our current results indicate a fast conflict detection process and suggest a complex and task-dependent conflict resolution process in FEF.

We trained the two animals using a highly similar paradigm (the same people trained them, we used the same setup, same procedures, same timings, etc.). Therefore, the difference between the animals might be caused by interindividual differences in task execution strategies (as human subjects would do). We do not know whether or how such interindividual differences developed during learning and training.

We prefer to refrain from speculating whether humans and monkeys would resolve the task in the same manner. Although it is a very interesting question and although we believe that the general neuronal mechanism might be highly similar. That said, it is very difficult to provide conclusive evidence for such statements.

4. The authors might consider discussing Di Bello et al., *Cerebral Cortex*, 2022. This study describes proactive and reactive distractor suppression mechanisms. Cosman et al. *Current Biol.* 2018 show that these suppressive signals arise in FEF before they arise in V1. These two studies are different from the present study in many ways, yet they also deal with the selection of relevant information and suppression of irrelevant information. Here, the authors show very fast conflict detection. Because conflict cannot arise reactively as it results from a learning process of cue significance, I would think that this encoding of conflict, although extremely fast, is of proactive nature, similar to the proactive suppression described by Di Bello et al. What are the thoughts of the authors on this? How much is overlearning contributing to the reported observations? Are the terms automatic and bottom-up driven most appropriate?

We thank the reviewer for this suggestion. We also agree with the reviewer that proactive suppression may be involved in conflict detection. We included a paragraph in the discussion section (lines 329-339) to discuss these studies.

“The former cells respond lower to congruent than incongruent cues, while the target-encoding neurons respond higher to congruent cues (Figure 3). The timing of conflict detection is consistent with previous results indicating that a target selection signal appears around 90ms after stimulus onset in a search task⁴⁵, suggesting shared mechanisms of conflict detection and target selection, especially with proactive suppression of distractors⁴⁵⁻⁴⁷. In other words, the distractor is proactively suppressed before it impairs target selection at both neuronal and behavioral levels in congruent compared to incongruent trials. Moreover, this process may contribute to the detection of the conflict. However, based on the current results only, it is difficult to fully understand the exact relationship between conflict processing and target selection.”

We think that learning and training are essential to the NCE. On the other hand, if the task requires subjects to find the target as soon as possible (e.g., in a reaction time task), we believe that the timing of conflict detection will not greatly differ from what we observed in the present study, while the latency of conflict resolution will probably be much faster. We agree that ‘automatic and bottom-up’ may be confusing, therefore, we deleted ‘bottom-up’ throughout the manuscript and changed the following sentences.

Line 24 in the abstract: “This suggests that conflict detection relies on a fast mechanism in FEF.”

Line 167-168: “Thus, conflicts are detected quickly in the FEF, as suggested by the short and small variations of NCE latencies.”

Line 320-322: “Our data suggest that the conflict is detected fast and independent of individuals differences and task rules, while its resolution is more complex and may depend on interindividual task-solving strategies and other cognitive processes (Figure 5).”

Line 394-396: “Independent of the underlying mechanisms, our results reveal that conflict resolution is a more complex process than conflict detection, and is affected by the task rules, interindividual differences, and possibly other cognitive processes.”

5. The authors perform a heavy selection on the neurons included in the present study (as described in detail in the material and methods section). While this is not a problem, it does raise in my mind the question of what the other cells are actually doing and how they are contributing -or not- to the computations described by the authors. I am not requesting additional analysis, but I think it would be useful to the readers to discuss this at some point.

We thank the reviewer for the suggestion. As indicated by the reviewer, we only included the visual and visuomotor neurons with a clear RF covering one of the two visual stimuli (the target or distractor) in our analysis. It is well known that the FEF houses at least three types of neurons based on their functions: visual neurons which only respond to visual stimuli in their RF; motor neurons responding before/during saccades; and the visuomotor neurons responding to both visual stimuli and

before/during saccades. In our study, we showed that the conflict modulates the visual response of visual neurons and/or visuomotor neurons -which we did not distinguish in our experiment. Nevertheless, we do not think that motor neurons or the motor response in FEF contribute to conflict processing in our task since we did not require the animals to make saccades. According to our hypothesis, in an experiment in which the subjects were required to make a saccade to the target, we speculate that if the saccade was directed towards the neuron's motor field, the motor neuron would be more active in congruent compared to incongruent trials. Conversely, if a saccade was directed away from the neurons' motor field, the neuron would exhibit higher responses during incongruent compared to congruent trials.

Inspired by the reviewer, we included the following sentences in the discussion (lines 397-410).

("Visual, visuomotor, and motor neurons have been identified in the FEF⁶⁶⁻⁷⁰. In the current study, we analyzed neurons that exhibited significant visual responses to peripheral stimuli (methods), therefore only visual and visuomotor neurons were considered. Since FEF neurons are typically not modulated by the planning and execution of manual operant behaviors⁷¹, and since our task did not involve saccades, we speculate that the motor components of the motor and visuomotor neurons may not exhibit conflict-related responses in our experiment. However, if a task would require subjects to make saccades, we regard it plausible that motor and visuomotor neurons would also show conflict-related responses. In that case we speculate that if a saccade would be directed towards the neuron's motor field, it would be more active in congruent compared to incongruent trials. Conversely, if a saccade would be directed away from the neurons' motor field, the neuron would exhibit higher responses during incongruent compared to congruent trials. Further studies, however, are needed to investigate the involvement of FEF's motor neurons in conflict processing when other operant behaviors than hand responses are required.")

Minor

1. Can you please indicate in figure 3efgh and corresponding S1 plots proportion of units in each bin with statistically different response between baseline and post cue response in interval of interest?

We have updated Figure 3 and Figure S1 accordingly.

In the original manuscript, the duration of the time windows we used to calculate the MIs in Figure 3E-H are different, i.e., 200ms (-200-0ms before the cue) for the baseline, and 300ms (100-400ms after the cue) for the response. In the current version, we used the same duration (200ms, 100-300ms after the cue for the response), therefore, the numbers slightly changed in Figure 3E-H compared to the original version. Although it doesn't change our conclusions at all, we think this analysis is slightly more optimal.

while at pop level, incongruency effects are clear, this is not the case at single cell level. Indeed although median of MI depart from 0, this is very subtle. How do you interpret this?

We thank the reviewer for the questions. The MIs in Figure 3E-H range from 0.05 to 0.1, corresponding to ~ 10% to 20% response changes, and we found that more than 40% of the neurons showed significant effects. Moreover, the results are consistent across subjects (Figure S4). Therefore, we consider this effect sizeable. However, as indicated by the reviewer, we found a variety of effect sizes (following a normal distribution).

Obviously, the population signals in FEF guide behavior and the response of a single neuron has, most likely, only limited impact.

How predictive is the modulation index of these distractor-neurons for spatial cue (cong or incongr) of the modulation index for color cue? In other words, is the distractor-population implementing spatial and color information along a mixed selectivity schema or along a schema of two independent populations. MI distributions speaks for former, which is what you would expect in the FEF.

We believe this question is already answered in our previous paper (Figure 6bd, Yao and Vanduffel 2022). Our data support the latter scenario. More than 90% of the neurons showed consistent effects across rules.

2. Figure 4, please show histograms for individual MIs in supplementary figure in order to back up reported results on median MI distributions on page 11.

We have updated the histograms in Figure S3.

Reviewer #2 (Remarks to the Author):

Review – Conflict resolution in macaque frontal eye fields

Yao and Vanduffel

The manuscript by Yao and Vanduffel describes the results of a study in which the authors sought to investigate the twin processes of conflict detection and resolution in the macaque FEF.

Two rhesus monkeys were each trained to perform two tasks – one in which the identity of a colour cue instructed the spatial location of a forthcoming target dimming, to which the animals responded with a button press. In this task, the color cues were presented at spatial locations either congruent or incongruent with respect to the location indicated by the color. In a second task, the spatial location of the cue indicated the spatial location of the target dimming, and the colour of the cue was irrelevant. In this case, congruent and incongruent were defined based on the relationship between the cue location and the direction instructed on their identities in the colour task. The authors report the presence of behavioral and neuronal congruency effects -(NCE) – increases in RTs and decreases in performance in incongruent vs congruent trials, and a neuronal NCE – the difference in discharge rates between congruent and incongruent trials when either the target or distractor was present in the neurons' response field. They additionally report the timecourses of the discharge rates in all experimental conditions for two populations

of FEF neurons – distractor encoding and target encoding. Generally speaking, activity for distractors was higher for incongruent than congruent trials, reflecting the presence of conflict. Conversely, for target-preferring neurons, activity was greater for targets during the cue period for congruent than incongruent trials, and this pattern subsequently reversed and then resolved during the delay period, which the authors ascribe to a cognitive process of conflict resolution. The authors thus argue that signals related both conflict detection and resolution are present in the FEF, and that their data are inconsistent with theories positing the ACC as a site of conflict detection, which is then signalled to prefrontal areas such as the FEF.

Overall, the task is well designed, and the animals behavior is well-characterized and consistent with a task engendering conflict. A sufficient number of trials and neuronal data were collected to answer the questions at hand and all analyses seem appropriate. I have some specific comments I would like address, listed below. I would be prepared to accept the manuscript if the following are addressed.

We thank the reviewer for the comments and summary of our study.

1. It seems as though the authors set up a bit of a straw man in the introduction with respect to the conflict monitoring (CM) hypothesis, which implicates the ACC in conflict detection and frontal areas in conflict resolution. In point of fact, while human fMRI studies have supported this hypothesis, there have not been any observations of conflict-related activity in ACC in studies recording single unit activity during tasks invoking conflict done more than 15 years ago (i.e. Nakamura, Ito). They do acknowledge these other studies in the discussion section but the introduction should be more inclusive of the facts and relative novelty here. They additionally seem to focus exclusively on ruling out the ACC as a site of conflict detection and based on prior work as well as their latency analysis, this seems plausible. They then, at least to my reading, seem to make the claim that conflict, in this case, is detected in the ACC and not elsewhere. Although some conflicting evidence does exist (i.e. Nakamura), prior work has found evidence for conflict signals in the SEF (Stuphorn et

al, 2000) which could be sent to the FEF with a latency consistent with their observations. Thus, conflict need not necessarily be detected specifically by the FEF but rather it could be receiving this signal from the SEF, after which the resolution process follows. This would also suggest that conflict signals in the FEF are not “bottom-up” as the authors argue in the discussion section. This should be acknowledged.

We thank the reviewer for the comments and we are sorry for the confusion of the message we aimed to deliver. In fact, whether or not ACC participates in conflict detection is not the point we aimed to make in the present manuscript. Instead, we focused on the temporal dynamics of conflict processing in FEF, with high temporal resolution and at neuronal level. We acknowledge that the introduction can be more elaborated as the reviewer suggested. To this end, we introduced the studies mentioned by the reviewer and profoundly reorganized the introduction. (line 49-71)

“We recently reported a profound conflict signal in neurons of the frontal eye fields (FEF) which we referred to as the neuronal congruency effect (NCE)¹². The NCE reflects the difference in neuronal response (at both single-cell and population levels) evoked by congruent and incongruent conditions. This signal provides a compelling neuronal mechanism for explaining the behavioral congruency effect.

Previous functional imaging and electrophysiological studies in humans and non-human primates suggest that fronto-parietal areas are involved in conflict processes^{3,11,13–31}. Yet, the temporal dynamics of conflict processing in the brain, including conflict detection and resolution, is poorly understood. Currently, the “conflict monitoring” (CM) model is the most prevalent theory to explain conflict processing. This theory suggests that a conflict is detected in medial frontal cortex (i.e., anterior cingulate cortex, ACC) after which this information is transmitted to lateral prefrontal cortex (LPFC) to initiate a conflict resolution process^{16,32–34}. This theory, however, has been challenged by other models suggesting that a conflict can be detected and resolved within LPFC without relying on a signal from ACC^{35–39}. Yet, conclusive neuronal evidence supporting either of these theories is limited due to the low temporal and/or spatial resolution of imaging techniques and the inconsistent results from single-unit recordings in ACC^{5,28,40–44}. Investigating the temporal dynamics of conflict

processing and target selection signals will be important to understand conflict processing in the brain, but also validate different theoretical and computational models. To this end, we analyzed in detail the time-courses of the NCE and target selection signals in the FEF to investigate how and when a conflict is detected and resolved in prefrontal cortex. Our study will help bridge the gap between abundant human imaging results and computational models related to conflict processing with single-neuron data from non-human primates.”

2. I am somewhat confused by the separate characterization of “distractor preferring” and “target preferring” populations here and wonder how these were established. To my mind, the design here would be to investigate the congruent and incongruent conditions for both tasks, with either the target and distractor inside the rf (i.e. as done in search tasks, where all conditions are compared within the populations of neurons). Since FEF neurons typically show initially indiscriminate responses to identical visual stimuli, unless there is some bias induced by training (Bichot et al., 1996), I wonder how this is the case. Or perhaps this is an artifact of the fact that the authors could not optimize the response fields for all neurons recorded. Some comment to clarify this would be appreciated.

The reviewer is correct that we actually compared different conflections within the same populations of neurons. We established the target- and distractor-encoding neurons based on the conditions. The rationale is that we would like to explain how FEF neurons respond when subjects are performing our task. We presented one target and one distractor stimulus on the screen in our task, therefore, in the brain areas (as FEF) with a retinotopic organization, there must be one group of neurons (which we labeled target-encoding neurons) with RFs covering the target, and another group of neurons (distractor-encoding neurons) covering the distractor. We did not record the two groups of neurons simultaneously (which requires recording in left and right FEF simultaneously). Instead, we manipulated the location of the target and distractor in different conditions to investigate the two groups of neurons. Obviously, this is very dynamic as distractor-encoding neurons become target-encoding neurons, simply

depending on the position of the target and distractor relative to the receptive fields of the neurons. We added one more sentence in the results section lines 110-113.

“Please note that in the current study, the target- and distractor-decoding neurons are the same group of neurons in different experimental conditions (i.e., TarIn: target inside RF, “target-decoding” neurons, solid lines in Figure 2C; DisIn: distractor inside RF, “distractor-encoding” neurons, dashed lines in Figure 2C).”

In the present study, we are interested in conflict detection and resolution. The NCE (which directly compares congruent and incongruent conditions as the reviewer indicated) is used to investigate conflict detection, while the target selection signal (which compares the target-in and distractor-in conditions in only the incongruent or high conflict trials) is used to investigate conflict resolution. Please note that, strictly speaking, we cannot directly investigate the process of conflict resolution, we can only investigate the consequence of the conflict resolution: i.e., whether the conflict is resolved or not. We assume that the conflict is resolved in hit trials. At neuronal level, the timing of conflict resolution is suggested by the time when a significant target selection signal appears in FEF, as shown in Figure 5.

We are rather confident about the relationship between the locations of the stimuli and the RFs of the neurons we studied. We are sure that one stimulus was presented in the RF while the other is not - by offline selection of the neurons (detailed in the methods). The PSTH of Figure 4 in our previous paper (Yao and Vanduffel, 2022, *Nature Communications*) confirms our argument. There was a clear response enhancement to target dimming when the target was presented inside the RFs (solid lines), which was absent when the target was presented outside the RFs (dashed lines). Figure 3A and Figure 4A in the current paper corroborate this finding: we found a significant and short-latency response enhancement induced by the cue when it was presented inside the RF vs outside.

As the reviewer suggested, the FEF neurons showed indiscriminate responses to identical visual stimuli before the color cue (Figure RR1) in our study. Nevertheless, we found significant conflict modulation during and after the cue, i.e., the NCE. This effect is our major finding and is described from different perspectives in our present

manuscript and our previous paper (Yao and Vanduffel, 2022). We agree with the reviewer that the NCE is related to learning and training. The NCE and the history and similarity effect described by Bichot *et al.* (1996) may share similar mechanisms. However, more studies are required to confirm this argument.

3. Have the authors investigated any sequential effects ? This might strengthen the findings of conflict signals and conflict resolution here if there are sufficient trials to do this. One might expect, for instance, that conflict would be highest for spatial trials immediately following a color trial, since they must now ignore color information and use the spatial location of the cues to facilitate detection of target dimming – especially in the incongruent. In this case, one might expect a higher NCE on these trials, relative to those on which the preceding trial was the spatial task. Just a suggestion but I would be interested to know if the authors have considered or tried this.

We acknowledge the reviewer for their valuable suggestions. We recognize the significance of investigating sequential effects within our dataset, as they provide important insights into cognitive flexibility and stability. In addition to exploring the rule-dependent sequential effect suggested by the reviewer, we are also considering sequential effects related to congruency and error. The sequential congruency effect, for example, is a highly relevant topic for understanding cognitive control across trials. Our ongoing analysis encompasses both neuronal and behavioral levels, which requires additional time. We believe that these analyses, focused on various sequential effects, will yield substantial data for another research paper. For the current paper, our analysis primarily centers on the within-trial dynamics and we believe that adding the analysis of sequential effects goes beyond the scope of the present manuscript .

4. The authors show evidence for behavioral and neuronal congruency effects (NCE). The implicit assumption here is that the NCE underlies the behavioral effect, but no relationship between the NCE and performance is described. Presumably the magnitude of the NCE in the last few hundred ms of the delay period reflects the extent

to which the earlier conflict in the task has been resolved, and spatial attention deployed to the target and away from the distractor location. What if the authors were to perform a median split of the data based on the distributions of the NCEs for all of the target and distractor neurons in an epoch consisting of the last few hundred ms of the delay period, and then compare the performance for the high NCE vs low NCE trials ? Such an analysis could be most illuminating with respect to the relationship between the extent of conflict resolution and performance and provide an explicit link between the neural and behavioral data.

We express our gratitude to the reviewer for their valuable suggestions. In fact, the questions raised by the reviewer was addressed in our previous paper (Yao and Vanduffel, 2022). In that study, we primarily focused on the neuronal response occurring 500ms prior to the target dimming and we investigated the correlation between the neuronal congruency effect (NCE) and performance. As suggested by the reviewer, we performed a similar analysis by categorizing the trials based on the reaction time duration into fast and slow trials. Our findings revealed that the amplitude of the NCE was not correlated with reaction times, as depicted in Figure 7 of our previous paper. Specifically, we observed a significant NCE in both fast and slow trials, while the reaction times between the congruent and incongruent trials did not always show significant differences. These results indicate that the NCE is a stable effect independent of reaction time, supporting the notion that it is an enduring phenomenon.

Minor points:

Line 93, performance reported for Monkey S does not match figure 2A.

We are sorry for the typo in the text, we have corrected the performance accuracy to 93.8% in the result (line 114).

We changed the sentence at lines 385-387 to be more precise:

“At the neuronal level, conflict resolution can be implied by the response difference between the target and distractor.”

REVIEWERS' COMMENTS:

Reviewer #1 (Remarks to the Author):

I have now analyzed the response of the authors to the comments of both reviewers. I am satisfied and do not have additional concerns. I thus would like to congratulate them on this work and I am looking forward to the upcoming study on the trial history effects.

Reviewer #2 (Remarks to the Author):

The authors have adequately addressed my concerns and comments and I would recommend the revised manuscript for publication.

REVIEWERS' COMMENTS:

Reviewer #1 (Remarks to the Author):

I have now analyzed the response of the authors to the comments of both reviewers. I am satisfied and do not have additional concerns. I thus would like to congratulate them on this work and I am looking forward to the upcoming study on the trial history effects.

Reviewer #2 (Remarks to the Author):

The authors have adequately addressed my concerns and comments and I would recommend the revised manuscript for publication.

Our sincere thanks go to the reviewers for their valuable critique, comments, and suggestions on this manuscript, which played a crucial role in shaping the final publication of our manuscript. We enjoyed the discussion of our work with the reviewers.